

# Improved meteorology and surface energy fluxes in mesoscale modelling using adjusted initial vertical soil moisture profiles

Igor Gómez[1,2], Vicente Caselles[1], María José Estrela[3], Juan Manuel Sánchez[4], Eva Rubio[5], Juan Javier Miró[1]

[1]Earth Physics and Thermodynamics Department, Faculty of Physics, University of Valencia, Doctor Moliner, 50, 46100 Burjassot, Valencia, Spain
[2]Environment and Earth Sciences Department, Faculty of Sciences, University of Alicante, Section 99, E-03080 Alicante, Spain
[3]Geography Department, Faculty of Geography and History, University of Valencia, Avda. Blasco Ibáñez, 28, 46010 Valencia, Spain
[4]Applied Physics Department, EPC and IDR, University of Castilla-La Mancha, Avda. España s/n, 02071 Albacete, Spain
[5]Applied Physics Department, University of Castilla-La Mancha, Avda. España s/n, 02071 Albacete, Spain

*Correspondence to*: Igor Gómez (Igor.Gomez@uv.es)

**Abstract.** The Regional Atmospheric Modeling System (RAMS) is being used for different and diverse purposes, ranging from atmospheric and dispersion of pollutants forecasting to agricultural meteorology and ecological modelling as well as for hydrological purposes, among others. The current paper presents a comprehensive assessment of the RAMS forecasts, comparing the results not only with observed standard surface meteorological variables, measured at FLUXNET stations and other portable and permanent weather stations located over the region of study, but also with non-standard observed variables, such as the surface energy fluxes, with the aim of evaluating the surface energy budget and its relation with a proper representation of standard observations and key physical processes for a wide range of applications. In this regard, RAMS is assessed against *in-situ* surface observations during a selected period within July 2011 over Eastern Spain. In addition, the simulation results are also compared with different surface remote sensing data derived from the Meteosat Second Generation (MSG) Spinning Enhanced Visible and Infrared Imager (SEVIRI) (MSG-SEVIRI) as well as the uncoupled Land Surface Models (LSM) Global Land Data Assimilation System (GLDAS). Both datasets complement the available *in-situ* observations and are used in the current study as the reference or ground truth when no observations are available on a selected location. Several sensitivity tests have been performed involving the initial soil moisture content, by adjusting this parameter in the vertical soil profile ranging from the most superficial soil layers to those located deeper underground. A refined adjustment of this parameter in the initialization of the model has shown to better represent the observed surface energy fluxes. The results obtained also show an improvement in the model forecasts found in previous studies in relation to standard observations, such as the air temperature and the moisture fields. Therefore, the application of a drier or wetter soil in distinct soil layers within the whole vertical soil profile has been found to be crucial in order to produce a better agreement between the simulation and the observations, thus reiterating the determining role of the initial soil moisture field in mesoscale modelling, but in this case considering the variation of this parameter vertically.



## 1 Introduction

Soil moisture has been found to be a key variable in the climate system, playing a fundamental role in the context of land surface energy and water budgets, through the total available energy partitioning between the sensible and latent heat fluxes (Dirmeyer et al., 2012; Gerken et al., 2015; Gallego-Elvira et al., 2016). In this regard, the land surface energy and water balances are related to the evapotranspiration term, which is strongly controlled by soil moisture over dry regions (Seneviratne et al., 2010). Thus, soil moisture influences both the air temperature and precipitation, and the variation of these

magnitudes may also affect the near-surface atmosphere and the structure of the atmospheric boundary layer.

Considering these key effects, an accurate initialization of the soil moisture parameter has been shown to have a positive impact in the simulation results produced by weather and climate models. This crucial significance of the soil moisture conditions on weather forecasts has been stated in a number of previous studies, both at short and medium range, and using different Numerical Weather Prediction (NWP) modelling environments (LeMone et al., 2007; van den Hurk et al.,

2008; Hong et al., 2009; Angevine et al., 2014; de Rosnay et al., 2014; Daniels et al., 2015; Daniels et al., 2016; Gómez et al., 2015b; Lin and Cheng, 2016; Dillon et al., 2016; Dirmeyer and Halder, 2016; Gómez et al., 2016b; Kalverla et al., 2016). In general, the initial field for soil moisture within NWP systems is based on reanalysis fields, such as the National Centers for Environmental Prediction (NCEP) Final FNL Operational Global Analysis dataset. On the other hand, satellite instruments can be used as well to observe land surface variable, such as ASCAT (METOP-A Advanced Scatterometer;

Bartalis et al., 2007), SMOS (Soil Moisture and Ocean Salinity; Kerr et al., 2010), or SMAP (Soil Moisture Active Passive; Entekhabi et al., 2010). However, satellite estimates generally correspond to the most superficial soil moisture layers, covering a depth of few centimeters (Dillon et al., 2016). For instance, SMOS and SMAP are mainly sensitive and representative of the first 5 cm of the top soil layer (Kerr et al., 2010; Entekhabi et al., 2010; Parrens et al., 2012; Leroux et al., 2016).

The main aim of the current study is to obtain a deeper insight of the influence of the initial soil moisture content on surface meteorology and energy fluxes forecasts. We use the Regional Atmospheric Modeling System (RAMS) mesoscale NWP environment for this purpose, bearing in mind the improvement of short to medium range weather forecasting of near surface variables (Gómez et al., 2014a,b,c; Gómez et al., 2015a). In the current study, this model is initialized using the heterogeneous land and soil parameters distribution provided by the NCEP FNL (Gómez et al., 2016c) dataset. Taking into

account that soil moisture plays a key role in land-atmosphere interactions (Betts, 2009; Seneviratne et al., 2010; Dirmeyer et al., 2012; Ferguson et al., 2012; Woldemichael et al., 2014; Gallego-Elvira et al., 2016; Lawston et al., 2017) as well as strongly controls surface turbulent fluxes in NWP models (Koster et al., 2009; Zaitchik et al., 2013; de Rosnay et al., 2014; Santanello et al., 2016), it is expected that a proper initialization of this soil parameter could produce reliable simulations. Following this reasoning, an adjusted initial soil moisture should produce an improvement of meteorological fields and

surface fluxes, especially over not densely vegetated locations. Considering these issues, we would like to answer the following questions: (1) What is the role of the initial soil moisture field provided by NCEP FNL over the area of study when



using RAMS, (2) Is it possible to obtain better results by means of an adjusted initial soil moisture content, (3) What is the influence of the soil moisture content applied over different vertical soil layers on the simulation results, that is, is it possible to improve the model results by customizing the soil moisture applied to different levels within the soil simulation profile.

Finally, and considering these three points, (4) how RAMS compares with other meteorological and atmospheric datasets, widely used by researchers and forecasters, such as remote sensing products as well as uncoupled Land Surface Models (LSM).

In order to answer the first three questions, we have designed and performed distinct sensitivity experiments drying the soil at different stages, with the aim of evaluating the impact of the soil moisture field in the forecast skill. In this sense,

the default RAMS run is that initialized using the NCEP FNL soil dataset. Using this information, we first estimate the effect of drying the soil only within the first uppermost layers in the soil model profile. This soil thickness is similar to and reproduces the one typically estimated in satellite missions, such as SMOS and SMAP, as seen before. Secondly, a drier soil moisture is applied to the uppermost layers as well as to the next deep soil layer. Finally, a drier environment is applied deeper underground, covering a depth of 25 cm. This experimental set-up can be achieved taking advantage of the flexibility

that offers the LSM implemented in RAMS, the Land-Ecosystem Atmosphere Feedback Model (LEAF; Walko et al. 2000), which represents the surface-atmosphere interaction processes, as it will be seen later. This LSM permits as well to customize the required layers in the soil profile on demand. Proceeding this way, we may evaluate not only the influence of the uppermost soil layers, as captured by remote sensing products, but also the influence of deeper soil layers, as provided by reanalysis and/or LSM models as well.

On the other hand, there is a critical need to produce skilful model-simulated meteorological and surface fluxes forecasts for applications and operations that rely on NWP models. Thus, it becomes essential to further evaluate the results produced by these models against observations and other data sources, in order to understand the origin of model limitations and strengths. In this sense and to answer the previously mentioned point (4), we have included surface remote sensing products derived from the Meteosat Second Generation (MSG) Spinning Enhanced Visible and Infrared Imager (SEVIRI)

(MSG-SEVIRI), and surface magnitudes obtained from the uncoupled Land Surface Model (LSM) Global Land Data Assimilation System (GLDAS; Rodell et al. 2004). Both surface variables datasets have been used so as to complement available *in-situ* observations when assessing the results produced by RAMS.

The paper is organised following this structure. Section 2 presents the methodology and datasets used, as well as a detailed description of the experimental design and the modelling strategy. Section 3 presents the simulation results and

discussion. Finally, conclusions are drawn in section 4.



## 2 Datasets and methodology

### 2.1 Model configurations

The selected forecasting period has been simulated based on the RAMS model (Cotton et al., 2003; Pielke, 2013), version 6.0, using three nested domains with horizontal resolution of 48 km, 12 km and 3 km, respectively. On the other hand, a total

of 45 vertical atmospheric levels are applied, with 22 levels included in the lowest 2,000 m and 8 levels in the lowest 300 m. Regarding physical parameterizations, the YSU PBL scheme (Hong et al., 2006; Gómez et al., 2016b), the Chen-Cotton scheme for longwave and shortwave radiation (Chen and Cotton, 1983), the Kain-Fritsch scheme for convection (Kain, 2004; Castro et al., 2002) and the Land-Ecosystem Atmosphere Feedback Model (LEAF-3; Walko et al. 2000) are used. RAMS is used to simulate the period from 6 to 12 July 2011. For each of these days, the model has been used in re-forecast

mode, performing a daily simulation with a forecast horizon of 36 h and a temporal resolution of 1 h, starting at 12 UTC the previous day. Thus, the first 12 h are left out as a spin-up period and only the corresponding complete day (the remaining 24 h) is considered in the evaluation. The NCEP FNL dataset at 6 h intervals and 1 x 1 degree resolution globally were used as initial and boundary conditions.

           We have designed a set of sensitivity experiments in order to evaluate the results. In this regard, four RAMS

simulations have been performed for each individual day within the period of study. The reference run is that provided by RAMS initialized using the FNL soil parameters (temperature and soil moisture) (EXP1). This is performed by means of the LEAF sub-model, version 3. LEAF-3 represents the surface energy budget, which partitions the net radiation into sensible, latent (evaporation plus transpiration), and soil heat fluxes. It incorporates the interactions between soil and vegetation, and their influence on each other and on the atmosphere at a subgrid scale (Walko et al., 2000). The soil model is used with a

total of 11 soil levels with higher resolution on the uppermost layers down to a depth 50 cm below the surface, including the following levels: 1, 3, 6, 9, 12, 16, 20, 25, 30, 40 and 50 cm. A second RAMS run (EXP2), manually reduces the soil moisture distribution in the surroundings of the area of study by multiplying this parameter with a factor of 0.5 (thus reducing the original soil moisture to the half), over the first three soil levels, that is, the first upper 6 cm. The third RAMS simulation (EXP3) applied this soil moisture reduction in the first upper 10 cm, while a fourth run (EXP4) uses the FNL soil

moisture reduced to the half over the first 25 cm underground. In all of these simulations, the remaining soil levels use the soil moisture directly provided by FNL. For instance, in the latter simulation (EXP4), the 50 % reduction in the initial soil moisture is applied to the first 8 soil levels, while the remaining three soil levels deeper underground use the original soil moisture values.

           An extensive description of the LEAF model can be consulted in Walko et al. (2000) and Pielke (2013). However,

we would like to include at this point some important issues of this model specially related to the soil moisture field. LEAF permits multiple surface types to coexist beneath a single grid-resolved column of air. Each surface type is then considered as a ''patch'' consisting of its own multiple soil, vegetation and canopy air layers as well as snow-cover (with the exception of water surface patches), as described in Walko et al. (2000). The surface fluxes are parameterized considering the





corresponding flux from ground surface to canopy air space, the flux from vegetation to canopy air space and the flux from
canopy air to the atmosphere. In the case of the latent heat flux, for example, the first two fluxes are represented by the next
equations, respectively:

$$\left(\lambda \mathrm{ET}\right)_{g,c} = \frac{C_p\left(q_g - q_c\right)}{r_d} \quad , \tag{1}$$

$$\left(\lambda \mathrm{ET}\right)_{v,c} = \rho\left(q_{vs} - q_c\right)\left[\frac{\left(\dfrac{W_f}{W_m}\right)^{2/3}}{r_b} + \frac{1 - \left(\dfrac{W_f}{W_m}\right)^{2/3}}{r_b + r_c}\right] \quad , \tag{2}$$

while the third one is parameterized including a term proportional to the frictional specific humidity. In equations (1) and (2),
$\rho$ represents the air density (kg m$^{-3}$) and $C_p$ is the specific heat capacity of air (J K$^{-1}$ kg$^{-1}$). In this sense, $\rho C_p$ represents the
volumetric heat capacity of air (J K$^{-1}$ m$^{-3}$), $q_g$ is the effective specific humidity at the surface (kg kg$^{-1}$), $q_c$ is the specific
humidity of the canopy air (kg kg$^{-1}$), and $q_{vs}$ is the saturated specific humidity at vegetation temperature (kg kg$^{-1}$), while $r_d$ is
the soil surface aerodynamic resistance (s m$^{-1}$) and $r_b$ is the bulk leaf boundary layer resistance (s m$^{-1}$). Finally, $W_f$ and $W_m$
are the water stored by vegetation (kg m$^{-2}$) and the maximum water reservoir capacity (kg m$^{-2}$), respectively, while $r_c$ is the
canopy resistance (s m$^{-1}$). Equation (2) represents the evapotranspiration rate from vegetation to canopy air, considering the
evaporation and the transpiration rates. More information about this parameterization can be found in Pielke (2013).

**2.2 Observational and modelling datasets**

Firstly, data from an anchor FLUXNET station, located over El Bonillo (BON), together with the measurements provided by
a portable weather station, located over Barrax (BRX), are used in the models' assessment. BON meteorological datasets
include hourly measures of 2-m temperature and relative humidity, 10-m wind speed and direction, surface sensible and
latent heat fluxes, and incident shortwave and longwave radiation. The portable weather station located over BRX measures
2-m temperature and relative humidity, 2-m wind speed, and incident shortwave and longwave radiation.

Even though no measurements of surface fluxes are directly measured over BRX, sensible and latent heat fluxes
derived from the STSEB (Simplified Two-Source Energy Balance) model are available for a small plot near the BRX
weather station, calculated within the framework of the experimental campaign described in Sánchez et al. (2014).

Thirdly, the uncoupled Land Surface Model (LSM) Global Land Data Assimilation System (GLDAS; Rodell et al.
2004) is also used. GLDAS provides derived products from the Noah uncoupled LSM, forced with observations and
uncoupled from an atmospheric model. The dataset used in the current study is the Noah LSM produced by GLDAS version
1 with a 3-hourly temporal resolution. This LSM model is based on 4 soil layers (0-0.1, 0.1-0.4, 0.4-1.0, 1.0-2.0 m) with a
horizontal spatial resolution of 0.25 x 0.25 degree globally. All meteorological magnitudes produced by Noah GLDAS, that
is, air temperature, relative humidity and wind speed, incident shortwave and longwave radiation, together with the surface
sensible and latent heat fluxes, as well as the soil moisture and skin temperature, are used for the models' assessment.



Finally, satellite-derived data from the Meteosat Second Generation (MSG) Spinning Enhanced Visible and Infrared Imager (SEVIRI) (MSG-SEVIRI) are also used in the current study. Land Surface Analysis Satellite Applications Facility

(LSA SAF), located at the Portuguese Meteorological Institute in Lisbon, provides MSG-derived high-level products, with a nominal resolution of 1 km (footprint of around 3 x 4.5 km in the study area), to a variety of user communities. From all the products provided by LSA SAF, the Downward Surface Shortwave Flux (DSSF; Brisson et al., 1999), the Downward Surface Longwave Flux (DSLF; Prata, 1996) and the Land Surface Temperature (LST; Caselles et al., 1997; Trigo et al., 2008) are used for the models' assessment. All these products are generated with spatial resolution and projection

corresponding to the characteristics of the MSG-SEVIRI instrument data.

## 3 Results and Discussion

### 3.1 Atmospheric conditions

We have selected the wind field as the starting point of the results analysis because we want to highlight here some relevant meteorological features that will be useful when tackling other meteorological variables. Thus, Fig. 1 shows the observed

wind field over BRX, where the dominant atmospheric conditions corresponding to the period of study are highlighted. In this figure, it can be seen that the main meteorological feature on 6 to 8 July is the presence of a Western synoptic advection, producing low atmospheric moisture over the study area, as it will be seen next, and high wind speeds (Fig. 1a). On the other hand, mesoscale circulations are developed on 9 and 10 July (Fig. 1b). In this case, a clear transition is observed between the day and night winds. As it will be seen below, a significant shift in the atmospheric moisture between night and day is

observed under these atmospheric conditions. Finally, the 11 July is characterised by the presence of an Eastern synoptic advection (Fig. 1c), while a Western synoptic advection is once again well established over the study area on 12 July (Fig. 1d). However, in this case some scattered cloudiness is observed.

Fig. 2 shows the comparison of the RAMS wind speed with the measurements recorded over BON and BRX. As similar results are obtained using the distinct RAMS experiments, we present the discussion using the results produced by

EXP3. In terms of the wind speed, RAMS properly reproduces the observations under mesoscale circulations (Fig. 2), with a MBE of -0.11 and 0.12 m s$^{-1}$ over BON and BRX, respectively, and RMSE of 0.8 m s$^{-1}$ in both cases (Table 1). However, RAMS yields higher wind speed values than those observed under the Western synoptic conditions at night-time (Fig. 2). In contrast, RAMS underestimates the observed wind speed during the day, considering these atmospheric conditions.

Considering the wind speed provided by GLDAS, Fig. 2 shows that this product reproduces rather well the

observations over BON and BRX. GLDAS is able to reproduce the main features observed for the whole simulation period, especially taking into account the daytime cycle. In contrast, it seems that GLDAS shows more difficulties at night-time, leading a more windy field than that simulated by RAMS. Confronting GLDAS with RAMS, a general overestimation is obtained for the GLDAS product in relation to the observations, as shown as well by the positive and larger MBE values




included in Table 1. For instance, a RAMS MBE of -0.08 raises to a GLDAS MBE of 2 m s$^{-1}$ in the case of BON, while the
RAMS MBE of 0.004 increases to 1.8 m s$^{-1}$ in the case of BRX.

**3.2 Surface fluxes**

A reduced soil moisture in the first soil levels produces a better representation of the sensible heat flux over BON (Fig. 3a),
especially under Western synoptic advections, where the MBE value changes from -50 W m$^{-2}$ using EXP1 to -30 W m$^{-2}$ using
EXP3 (Table 2). Focusing on this weather station (Fig. 3b), EXP1 shows a clear tendency to overestimate the observed latent
heat flux, as displayed by positive values of MBE, with a global MBE of 70 W m$^{-2}$ and an RMSE of 90 W m$^{-2}$ (Table 2).
Although the differences between the observations and the model are notably reduced by EXP3 in comparison to EXP1 over
BON, the overestimation of the latent heat flux is still maintained using EXP3. However, the reduction in soil moisture
applied to this simulation leads to a better representation of the sensible heat flux, reducing the total MBE in around 20 W m$^{-2}$
and the RMSE in 10 W m$^{-2}$. In terms of the latent heat flux, drying off deeper soil levels, such as in EXP4, produces a better
agreement between the model and the observations over BON. This RAMS simulation shows MBE of 4 W m$^{-2}$ and 3 W m$^{-2}$
and RMSE of 13 W m$^{-2}$ under mesoscale circulations and the Eastern synoptic advection, respectively, significantly
improving the results obtained by the other RAMS configurations.

Focusing on BRX station, RAMS tends to overestimate the results produced by the STSEB model of the sensible
heat flux, with differences higher than 100 W m$^{-2}$ (Fig 4c). Considering this weather station, a reduction in the soil moisture
field, such as that imposed in EXP3 and EXP4 increases the differences between the simulation and the STSEB results. This
is also reflected in the latent heat flux, where the best simulation is that obtained with the EXP1 run. It seems that this
simulation is properly capturing the high moisture content available over this area within the period of study, related to a
well irrigated plot which leads to large values of latent heat flux together with very small values of sensible heat fluxes, as
observed in Fig. 3c,d.

Contrasting the RAMS surface fluxes with those derived from the GLDAS product, a better agreement is obtained
when a reduced soil moisture is applied to RAMS. Confronting this GLDAS product based on the Noah LSM model with the
*in-situ* observations over BON, GLDAS results are in between EXP3 and EXP4 for both the sensible an latent heat fluxes
(Fig. 3a,b). There is a general tendency to overestimate the observed sensible heat flux, as showed by the positive MBE
score included in Table 2. This trend varies according to the dominant atmospheric condition, with values of 6 W m$^{-2}$ under
Western synoptic advections to values up to 40 W m$^{-2}$ under mesoscale circulations. On the other hand, GLDAS shows the
lowest correlation coefficient for the distinct atmospheric situations when compared to RAMS for the sensible heat flux.

Considering the latent heat flux over BON, GLDAS overestimates the observations as well, as indicated by the
global positive values of 19 W m$^{-2}$ (Table 2). Comparing the GLDAS results with those produced by RAMS for this
magnitude over BON, better results than EXP1, EXP2 and EXP3 are obtained in terms of MBE and RMSE, even though the
correlation coefficient is notably reduced for the GLDAS product. It is necessary to reduce the original soil moisture in




deeper soil levels, such as in EXP4, to obtain better representation of the latent heat flux over BRX and produce better results than GLDAS over BON.

On the other hand, comparing the EXP3 with EXP2 simulated turbulent fluxes, similar results are obtained for both the sensible and the latent heat flux (Fig. 4a,b). In the first case, slightly higher values are simulated by EXP3, while the
opposite is observed for the latent heat flux. This result should be expected as more moisture is removed from the soil in EXP3, which is translated in the surface fluxes outcomes.

Larger differences are observed over BRX considering the GLDAS product in terms of the turbulent fluxes (Fig. 3c,d). This could be anticipated to some extend as the STSEB fluxes displayed in this location correspond to a very small plot, and the GLDAS resolution is not likely to properly capture the distinctive features of this area, especially the irrigation
applied in the mentioned small plot. It seems that the working resolution of the GLDAS product is too coarse to properly capture the special features of BRX within the period of study.

### 3.3 Temperature and Moisture

Regarding the 2-m temperature at daytime (Fig. 5), EXP1 produces an underestimation of this magnitude for the whole simulation period, as indicated by the negative MBE values included in Table 3, with a global MBE of -1.5 and a RMSE of 2
ºC. Considering this magnitude during the night, more differences appear in the model's trend in relation to the observation, depending on the general atmospheric situation. In this sense, EXP1 properly simulates this magnitude on 6 to 8 July 2011 over BON, under a well established Western synoptic advection, with a minimum temperature difference between the observations and the simulations results of 0.10 and 0.9 ºC on 7 and 8 July 2011 (Table 4), while it is underestimated on 9 to 11 July, coinciding with mesoscale circulations and an Eastern synoptic advection, with values around -1.7 and -0.9 ºC,
respectively. Although the night-time temperature is well captured by RAMS under mesoscale circulations, it is overestimated under the synoptic advections over BRX, but with different degree of agreement, ranging from 2 to 5 ºC in the minimum temperature difference between the observations and the simulations results (Table 5). In the case of the maximum temperatures, EXP1 shows negative differences for the maximum temperature between the simulation results and the observations (similar to BON), as displayed in Fig. 5 as well.

A reduction in the initial soil moisture content in the upper soil levels leads to a reduced difference between the observations and the simulated results. For instance, EXP3 produces a different tendency over BON and BRX, producing in general higher global errors over BRX (Table 3). However, considering the maximum temperatures differences between the simulated results and the observations over these two weather stations (Tables 5 and 6), the general underestimation of EXP3 over BON, with values around -1.1 ºC as a mean value, is reversed in sing over BRX at the same time that the maximum
temperatures differences are reduced to values around 0.5 ºC. These differences are reduced as well over BON further drying the soil deeper underground, such as in EXP4. In this case, the maximum temperature differences are notably reduced, with a mean value of 0.2 ºC in contrast to the mean value of -1.1 ºC produced by EXP3. However, EXP4 tends to overestimate the



maximum temperature difference over BRX for the whole simulation period, with a mean value of 2 ºC. These differences between the observations and the RAMS results are also clearly seen in Fig. 5a,c.

If we focus on the air temperature provided by the GLDAS product (Fig. 5), maximum temperatures are really well captured in general, with a trend to underestimate the observations over BON, with a mean difference between the simulated results and the observations of -1.1 ºC (Table 4), while the mean maximum temperature difference is 1.5 ºC over BRX (Table 5). Considering the whole simulation period, a global mean bias of 0.04 ºC is obtained over BON, while the MBE raises to 1.9 ºC over BRX. The global MBE values (Table 3) conceals relevant differences between the observations and GLDAS 2-m

temperatures at night and day-time. In this regard, Table 4 and 6 show more differences at night-time, with a general overestimation of the air temperature within the whole simulation period, as clearly displayed by Fig. 5a,c as well. It seems that the higher horizontal resolution of GLDAS is not suitable to reproduce the temperature observations at night-time.

In terms of the 2-m relative humidity, values around 50 % or even lower are found in general at night under the Western synoptic advection on 6 to 8 July 2011, while they reach values greater than 90 % under mesoscale circulations on 9

and 10 July 2011. RAMS reproduces this magnitude properly over BON (Fig. 5b), but higher differences are obtained under mesoscale circulations over BRX (Fig. 3d). In this regard, although similar results are found over BON and BRX under the Western synoptic advections, more differences are obtained between these two weather stations under Eastern advections, such as the Eastern synoptic advection and mesoscale circulations. In both cases, higher MBE and RMSE are simulated by RAMS over BRX, with values around -20 % and -30 %, respectively, highlighting a clear underestimation of the observed

relative humidity (Table 6). These differences are related to distinct dominant wind flux conditions over each station location (not shown). In this regard, even though the general Eastern advections on 9 to 11 July are established over BRX, this is not the case for BON, where Western winds dominate, leading to a dry environment with lower relative humidities than in the case of BRX.

Contrasting the EXP3 with EXP2 simulations for the 2-m temperature and relative humidity, similar results are

obtained for both magnitudes (Fig. 4c,d). Drying the soil, as in EXP3, leads to slightly higher sensible heat fluxes and lower latent heat fluxes (Fig. 4a,b). However, these slight differences in the surface fluxes do not lead to a significant drier and warmer environment, and really similar results are obtained for EXP2 and EXP3 in terms of the thermodynamic variables, which is also clearly shown by the MBE and RMSE in tables 4 and 7.

On the other hand, the air relative humidity provided by GLDAS produces results closer to RAMS during the day,

especially considering EXP3 and EXP4 (Fig. 5b,d). In general, GLDAS produces a lower relative humidity than RAMS at night, practically for the whole simulation period, which is more notably under mesoscale circulations over both BON and BRX. However, GLDAS is able to properly capture this magnitude when a synoptic advection is established as the main meteorological feature over the area of study. This is particularly true under the Eastern synoptic advection observed on 11 July (Table 6), with a MBE of -1.0 and -19 %, and an RMSE of 7 and 20 %, over BON and BRX, respectively. Similar

results are found under the cloudy Western advection on 12 July, with MBE of -0.5 and -8 %, and an RMSE of 5 and 10 %, over BON and BRX, respectively.





Considering the meteorological variables and the surface fluxes, there is a clear connection between the 2-m temperature and the turbulent fluxes, as shown in Figs. 4 and 6. In this regard, the general cold bias produced by RAMS in relation to the observations is related to an underestimation of the sensible heat flux and an overestimation of the latent heat

flux. Following this argument, a proper representation of the initial soil moisture field produces notably reduced differences between RAMS and the observations in terms of the 2-m temperature, related to a decrease in the simulated latent heat flux and an increased sensible heat flux. Thus, it would be necessary to increase the sensible heat flux and decrease the latent heat flux so as to reduce the gap between RAMS modelling results and the observations, such as in EXP4 over BON and EXP3 over BRX. This can be achieved by drying different soil levels compared to the high original soil moisture content provided

by FNL. Furthermore, this lower soil moisture content is supported by the GLDAS values.

### 3.4 Radiation Components

Looking at the incident shortwave radiation field (Fig. 6a,c), little differences are observed between the distinct RAMS simulations, with all of them capturing really well the observations for practically the whole simulation period. MBE around 0.8 to 8 W m$^{-2}$ are obtained under Western synoptic advections and mesoscale circulations over BON (Table 7). On the other

hand, MBE raises to 30 W m$^{-2}$ under the Eastern synoptic advection. Considering the 12 July, the observations of the downward shortwave radiation show some persistent clouds over BON throughout the day. Satellite images of cloudiness distribution (not shown) confirm the persistence of these scatter clouds over the area of study.

To evaluate the radiation components, in addition to the *in-situ* observations, the MSG-SEVIRI DSSF product is also used to be compared with the EXP3 run. Besides, the GLDAS product is also included in the analysis. RAMS adjusts

the results provided by both GLDAS and SEVIRI in terms of the incident shortwave radiation, reducing the MBE under the Western synoptic advection. Table 7 shows that EXP3 performs better than SEVIRI and GLDAS under this atmospheric conditions as well as under mesoscale circulations. However, larger differences are found under the Eastern synoptic advection and the cloudy Western advection (Table 7), especially in this second case. Nevertheless, considering each product independently, higher MBE and RMSE and lower correlation coefficients are obtained under both the Eastern and the cloudy

Western synoptic advections. The differences obtained between RAMS and the measurements as well as between this model and the DSSF product under cloudy conditions agree with those found in previous studies performed over the Western Mediterranean coast (Gómez et al., 2016a; Federico et al., 2017). It is highlighted here the remarkable difficulty to forecast cloudiness when using mesoscale modeling, especially where scatter clouds are present over the area of study. It seems that the extension of cloudiness, even though well captured over BON, is more extended than really observed, based on the

satellite images (not shown). Additionally, the DSSF product has also been found to produce larger errors under cloudy conditions compared to those obtained for clear skies (Cristóbal et al., 2013).

On the other hand, the incoming longwave radiation is underestimated by GLDAS, as in the case of RAMS and SEVIRI. Although this underestimation is the general trend obtained over both BON and BRX (Fig. 6b,d), the modelling results adjust better to the observations over BON using SEVIRI-DSLF, with a global MBE value of 5 W m$^{-2}$ and an RMSE





of 10 W m⁻², in contrast to the RMSE of 30 and 16 W m⁻², obtained by EXP3 and GLDAS, respectively (Table 8). In general, the greatest differences between the modelling results and the observations of this magnitude are obtained using RAMS, as mentioned above in relation to the different statistical scores shown in Table 8. Considering this model, little discrepancies are found varying the initial soil conditions (not shown). This mentioned difficulty of both RAMS to reproduce the longwave downwelling flux, could be related to a deficiency in the implemented radiation scheme and seems to be the most likely

reason for the model biases in this magnitude.

Finally, Fig. 7 shows a comparison of the outgoing longwave radiation simulated by the different RAMS configurations with the BRX observations. The MBE and RMSE increase for EXP4 (Table 9), producing slightly greater daytime outgoing longwave radiation, with EXP1 closer to the observations. The general trend at night points towards an overestimation of the observations in all RAMS simulations (Fig. 7a). In this regard, comparing EXP3 with EXP2 (Fig. 7b),

no significant differences are found between these two RAMS configurations for the whole simulation period.

### 3.5 Surface Temperature and Moisture

Contrasting the RAMS LST field with that provided by SEVIRI and GLDAS, Fig. 8a,c reflect that RAMS and GLDAS have a lower diurnal temperature range than that provided by SEVIRI. Considering the night-time temperatures, RAMS produces a general overestimation of the SEVIRI LST reference field, with values closer to those provided by GLDAS. This

overestimation of the SEVIRI LST field is clear under synoptic advections, while the differences between the modelling results and the satellite derived LSTs are reduced under mesoscale circulations. However, the trend obtained in RAMS in relation to SEVIRI is reversed at daytime over BON, where all simulations produce lower LST values than SEVIRI. This underestimation is also highlighted contrasting the GLDAS LST outputs to those simulated by RAMS.

Finally, if we focus on the soil moisture field simulated over BON and BRX for the uppermost soil layer (Fig. 8b,d),

EXP1 shows a clear tendency to produce larger values than the other simulations. Although no soil moisture measurements are available over the corresponding weather stations, GLDAS can also be used as a reference field for this magnitude. Fig. 8b,d show a reduced difference, of about 0.02 m³ m⁻³, between GLDAS and EXP3 and EXP4, for both BON and BRX. In this case, GLDAS is closer to RAMS, where GLDAS shows soil moisture values around 0.10 m³ m⁻³, while EXP3 shows values around 0.08 m³ m⁻³ or slightly higher over BRX on 6 to 8 July.

Finally, Fig. 9 presents the soil moisture field corresponding to the most four upper levels simulated by EXP3 and EXP4 over BON (Fig. 9a,c, respectively) and BRX (Fig. 9b,d, respectively). Soil_L1 is located at 2 cm, Soil_L2 at 4.5 cm, Soil_L3 at 7.5 cm and Soil_L4 at 10.5 cm. GLDAS soil moisture corresponding to the first soil layer of this model (0-10 cm) is also included in Fig. 9 as a reference. The RAMS results obtained at this two weather station locations is rather similar. The effect of drying the soil deeper underground in contrast to just drying the uppermost levels is clear in this figure.



## 4 Summary and conclusions

The current study evaluates the ability of the RAMS atmospheric model to forecast different meteorological variables and surface energy fluxes over a region in Eastern Spain, and the sensitivity of the model results to distinct soil moisture conditions. These soil configurations are set based on drying different soil levels in the vertical soil profile taking into account the original soil moisture content provided by the FNL soil moisture dataset, used for RAMS initialization. The response of the model is evaluated by progressively drying deeper soil levels from the uppermost soil levels up to the first 25 cm in the soil model configuration. In order to perform a comparative assessment of the model results, in addition to available *in-situ* observations, we have used remote sensing data derived from the MSG-SEVIRI sensor and the uncoupled LSM GLDAS. Initializing RAMS with the soil moisture provided by FNL shows a general positive bias in the latent heat flux over the area of study, which seems to be related to a too moist lower boundary layer. This damp environment leads to a negative bias in the sensible heat flux and cold temperatures when compared to the observations. Imposing a drier soil environment just in the upper soil levels, that is, the upper 6 cm, leads to a better adjustment between the model simulated magnitudes and the observations. This is the case over BRX, where continually drying the soil in deeper levels up to 25 cm produces worse results than using this dry soil in the upper levels. However, this is not the case over BON, where it is necessary to impose a drier soil deeper underground so as to obtain a better representation of the observed meteorology and surface turbulent fluxes. As a result, a drier environment is obtained and the simulated moisture field best fits the observations. Furthermore, the air temperature field is really improved in relation to the observations as well as the turbulent fluxes. A reduction in the initial soil moisture field produces a suitable agreement with the soil moisture produced by GLDAS, and reduces the differences between the observations and the modelling results.

A proper representation of the soil moisture field is then crucial in order to obtain a suitable representation of the surface fluxes. This has been highlighted in the current paper by means of a comparison between two very distinct moisture regimes: the dry environment observed over BON and the moister environment over BRX. In this regard, it seems that the soil moisture provided by FNL is too moist over the area of study, leading to a general overestimation of the latent heat flux and an underestimation of the sensible heat flux (as simulated by EXP1). However, the higher soil moisture values supplied by FNL are more suitable over BRX, considering the well-irrigated conditions of a small plot in this location during the simulation period, and when the STSEB model is applied.

These modelling results obtained in the surface fluxes are translated to the temperature and moisture fields as well. Therefore, it is worth noting the influence of the initial soil moisture content on the model results depending on the specific place. Thus, this parameter should be first considered when configuring a mesoscale model, taking into account as well the importance of heterogeneity, as we have seen that a drier environment is establish over BON but not over BRX for the same simulation period. Considering the vertical distribution of this parameter for the corresponding geographical location, not only near the surface but also deeper underground, is essential in order to improve and properly reproduce the observed meteorology and surface fluxes, as it has been demonstrated in the current study.



**Acknowledgement**

This work has been funded by the Regional Government of Valencia through the project PROMETEOII/2014/086 and by the
Spanish Ministerio de Economía y Competitividad and the European Regional Development Fund (FEDER) through the
project CGL2015-64268-R (MINECO/FEDER,UE). NCEP is acknowledged for providing the FNL analysis data for RAMS
initialization. National Centers for Environmental Prediction/National Weather Service/NOAA/U.S. Department of
Commerce (2000): NCEP FNL Operational Model Global Tropospheric Analyses, continuing from July 1999. Research Data
Archive at the National Center for Atmospheric Research, Computational and Information Systems Laboratory. Dataset.
http://rda.ucar.edu/datasets/ds083.2. Accessed 25 August 2014. The GLDAS data used in this study were acquired as part of
the mission of NASA's Earth Science Division and archived and distributed by the Goddard Earth Sciences (GES) Data and
Information Services Center (DISC) are acknowledged by disseminating the GLDAS data. Land Surface Analysis Satellite
Application Facility (LSA SAF) are acknowledged as well for providing the satellite data. The Spanish Ministry of Economy
and Competitiveness with co-funding from the European Development Regional Fund (MINECO/FEDER, UE; Project
AGL2014-55658-R, FORESTRENGTH), and the Education, Culture and Sports Department of the Castilla-La Mancha
Regional Council with co-funding from the European Development Regional Fund (FEDER) (Project PEIC-2014- 002-P,
ECOFLUX III) are acknowledged as well for funding the acquisition of surface fluxes data for RAMS validation.

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



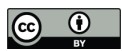

**Table 1. Correlation coefficient (R), Mean Bias Error (MBE; m s⁻¹) and Root Mean Square Error (RMSE; m s⁻¹) for RAMS-EXP3 and GLDAS 10-m wind speed over BON and BRX, considering the distinct dominant atmospheric conditions over the study area.**

| | BON | | | BRX | | |
|---|---|---|---|---|---|---|
| | R | MBE | RMSE | R | MBE | RMSE |
| Western synoptic advection | | | | | | |
| EXP3 | 0.616 | -0.08 | 1.7 | 0.493 | 0.004 | 1.6 |
| GLDAS | 0.905 | 2 | 2 | 0.852 | 1.8 | 2 |
| Mesoscale circulation | | | | | | |
| EXP3 | 0.837 | -0.11 | 0.8 | 0.553 | 0.12 | 0.8 |
| GLDAS | 0.874 | 1.3 | 1.5 | 0.475 | 0.9 | 1.5 |
| Eastern synoptic advection | | | | | | |
| EXP3 | 0.822 | 0.6 | 1.1 | 0.156 | 0.6 | 1.1 |
| GLDAS | 0.809 | 0.7 | 1.4 | 0.418 | 0.3 | 1.2 |
| Western synoptic advection (Cloudy) | | | | | | |
| EXP3 | 0.902 | -0.4 | 1.1 | 0.266 | -0.8 | 2 |
| GLDAS | 0.944 | 1.6 | 1.7 | 0.932 | 1.0 | 1.3 |
| All atmospheric conditions | | | | | | |
| EXP3 | 0.735 | -0.03 | 1.3 | 0.544 | 0.0015 | 1.5 |
| GLDAS | 0.863 | 1.6 | 1.9 | 0.792 | 1.2 | 1.7 |








**Table 2. Correlation coefficient (R), Mean Bias Error (MBE; W m⁻²) and Root Mean Square Error (RMSE; W m⁻²) for RAMS experiments and GLDAS surface fluxes (sensible heat flux, H; and latent heat flux, LE) over BON weather station, considering the distinct dominant atmospheric conditions over the study area.**

| | H | | | LE | | |
|---|---|---|---|---|---|---|
| | R | MBE | RMSE | R | MBE | RMSE |
| | Western synoptic advection | | | | | |
| EXP1 | 0.979 | -50 | 60 | 0.797 | 70 | 90 |
| EXP2 | 0.977 | -40 | 50 | 0.795 | 40 | 50 |
| EXP3 | 0.977 | -30 | 50 | 0.796 | 30 | 50 |
| EXP4 | 0.980 | 7 | 60 | 0.787 | -1.5 | 20 |
| GLDAS | 0.946 | 6 | 60 | 0.715 | 19 | 30 |
| | Mesoscale circulation | | | | | |
| EXP1 | 0.966 | -30 | 50 | 0.902 | 60 | 100 |
| EXP2 | 0.964 | -15 | 40 | 0.907 | 40 | 60 |
| EXP3 | 0.965 | -8 | 40 | 0.907 | 30 | 50 |
| EXP4 | 0.972 | 30 | 70 | 0.882 | 4 | 13 |
| GLDAS | 0.956 | 40 | 80 | 0.799 | 19 | 30 |
| | Eastern synoptic advection | | | | | |
| EXP1 | 0.973 | -15 | 40 | 0.951 | 70 | 100 |
| EXP2 | 0.971 | -1.5 | 50 | 0.962 | 40 | 60 |
| EXP3 | 0.972 | 10 | 60 | 0.953 | 40 | 50 |
| EXP4 | 0.977 | 50 | 100 | 0.878 | 3 | 13 |
| GLDAS | 0.955 | 30 | 70 | 0.745 | 20 | 40 |
| | Western synoptic advection (Cloudy) | | | | | |
| EXP1 | 0.848 | -80 | 120 | 0.405 | 40 | 80 |
| EXP2 | 0.864 | -50 | 100 | 0.514 | 30 | 60 |
| EXP3 | 0.863 | -50 | 100 | 0.479 | 20 | 50 |
| EXP4 | 0.836 | -18 | 110 | 0.432 | -0.8 | 30 |
| GLDAS | 0.846 | 13 | 100 | -0.177 | 10 | 50 |
| | All atmospheric conditions | | | | | |
| EXP1 | 0.943 | -40 | 70 | 0.727 | 60 | 90 |
| EXP2 | 0.948 | -30 | 60 | 0.749 | 40 | 60 |
| EXP3 | 0.947 | -19 | 60 | 0.744 | 30 | 50 |
| EXP4 | 0.948 | 16 | 70 | 0.728 | 0.8 | 20 |
| GLDAS | 0.928 | 20 | 70 | 0.608 | 19 | 40 |




**Table 3. Correlation coefficient (R), Mean Bias Error (MBE; ºC) and Root Mean Square Error (RMSE; ºC) for RAMS experiments and GLDAS 2-m temperature over BON and BRX, considering the distinct dominant atmospheric conditions over the study area.**

| | BON | | | BRX | | |
|---|---|---|---|---|---|---|
| | R | MBE | RMSE | R | MBE | RMSE |
| Western synoptic advection | | | | | | |
| EXP1 | 0.961 | -1.1 | 2 | 0.953 | 0.5 | 2 |
| EXP2 | 0.958 | -0.4 | 1.8 | 0.945 | 1.7 | 3 |
| EXP3 | 0.960 | -0.3 | 1.7 | 0.948 | 1.8 | 3 |
| EXP4 | 0.961 | 0.5 | 1.7 | 0.956 | 3 | 3 |
| GLDAS | 0.960 | 0.13 | 1.6 | 0.960 | 1.6 | 2 |
| Mesoscale circulation | | | | | | |
| EXP1 | 0.987 | -2 | 2 | 0.984 | 0.5 | 1.6 |
| EXP2 | 0.986 | -1.6 | 2 | 0.983 | 1.6 | 2 |
| EXP3 | 0.986 | -1.7 | 2 | 0.982 | 1.6 | 2 |
| EXP4 | 0.986 | -0.8 | 1.8 | 0.980 | 2 | 3 |
| GLDAS | 0.972 | 0.4 | 2 | 0.934 | 3 | 4 |
| Eastern synoptic advection | | | | | | |
| EXP1 | 0.982 | -1.6 | 1.9 | 0.991 | 1.9 | 2 |
| EXP2 | 0.984 | -0.9 | 1.5 | 0.981 | 3 | 3 |
| EXP3 | 0.985 | -0.8 | 1.5 | 0.982 | 3 | 3 |
| EXP4 | 0.988 | -0.03 | 1.4 | 0.966 | 4 | 4 |
| GLDAS | 0.955 | -0.16 | 1.9 | 0.952 | 2 | 3 |
| Western synoptic advection (Cloudy) | | | | | | |
| EXP1 | 0.843 | -1.2 | 3 | 0.731 | -0.7 | 3 |
| EXP2 | 0.880 | 0.07 | 2 | 0.876 | 0.9 | 2 |
| EXP3 | 0.886 | 0.2 | 2 | 0.873 | 1.0 | 2 |
| EXP4 | 0.902 | 1.2 | 2 | 0.880 | 1.8 | 3 |
| GLDAS | 0.961 | -0.7 | 1.3 | 0.955 | 0.12 | 1.2 |
| All atmospheric conditions | | | | | | |
| EXP1 | 0.963 | -1.5 | 2 | 0.946 | 0.5 | 2 |
| EXP2 | 0.961 | -0.8 | 1.8 | 0.952 | 1.7 | 3 |
| EXP3 | 0.962 | -0.7 | 1.8 | 0.953 | 1.8 | 3 |
| EXP4 | 0.964 | 0.17 | 1.8 | 0.956 | 3 | 3 |
| GLDAS | 0.963 | 0.04 | 1.7 | 0.933 | 1.9 | 3 |






**Table 4. Difference between the simulated and observed maximum and minimum 2-m temperature (ºC), for the distinct RAMS simulations and the GLDAS product, over BON weather station. For GLDAS, the maximum temperature is that obtained at 15 UTC, while the minimum temperature is the one obtained at 06 UTC.**

| Day | EXP1 | EXP2 | EXP3 | EXP4 | GLDAS |
|---|---|---|---|---|---|
| 06/07/2011 | -2 / 1.2 | -2 / 1.9 | -1.9 / 1.6 | -0.6 / 1.8 | -1.0 / 4 |
| 07/07/2011 | -1.9 / 0.10 | -1.1 / 0.8 | -0.7 / 0.5 | 0.6 / 0.6 | -0.5 / 3 |
| 08/07/2011 | -3 / 0.9 | -1.7 / 1.5 | -1.4 / 1.1 | -0.03 / 1.3 | -0.09 / 1.7 |
| 09/07/2011 | -3 / -1.7 | -1.7 / -1.5 | -1.3 / -2 | 0.07 / -1.9 | -0.8 / 3 |
| 10/07/2011 | -3 / -1.8 | -2 / -1.8 | -2 / -2 | -0.5 / -2 | -0.5 / 2 |
| 11/07/2011 | -1.8 / -0.9 | -0.6 / -0.9 | 0.018 / -1.4 | 1.2 / -1.5 | 0.15 / 3 |
| 12/07/2011 | -3 / 3 | -1.0 / 3 | -0.7 / 3 | 0.8 / 4 | -0.7 / -0.9 |
| Mean | -3 / 0.11 | -1.4 / 0.4 | -1.1 / 0.11 | 0.2 / 0.3 | -1.1 / 2 |


**Table 5. Same as Table 4, but over BRX weather station.**

| Day | EXP1 | EXP2 | EXP3 | EXP4 | GLDAS |
|---|---|---|---|---|---|
| 06/07/2011 | -1.7 / 2 | -0.3 / 4 | 0.05 / 4 | 1.4 / 5 | 0.3 / 3 |
| 07/07/2011 | -0.9 / 4 | 0.2 / 6 | 0.6 / 5 | 1.8 / 6 | 0.4 / 4 |
| 08/07/2011 | -1.1 / 5 | 0.06 / 6 | 0.5 / 6 | 1.9 / 6 | 1.6 / 1.8 |
| 09/07/2011 | -1.3 / 1.5 | 0.2 / 4 | 0.6 / 3 | 2 / 2 | 2 / 7 |
| 10/07/2011 | -1.6 / -0.3 | 0.002 / 0.3 | 0.6 / -0.5 | 1.7 / -0.9 | 2 / 4 |
| 11/07/2011 | 1.6 / 3 | 3 / 4 | 4 / 4 | 4 / 4 | 3 / 3 |
| 12/07/2011 | -1.9 / 2 | 0.14 / 4 | 0.6 / 3 | 2 / 4 | 1.5 / 1.0 |
| Mean | -1.0 / 2 | 0.5 / 4 | 1.0 / 4 | 2 / 4 | 1.5 / 3 |







**Table 6. Same as Table 3, but for the 2-m relative humidity (%).**

| | BON | | | BRX | | |
|---|---|---|---|---|---|---|
| | R | MBE | RMSE | R | MBE | RMSE |
| Western synoptic advection | | | | | | |
| EXP1 | 0.912 | 0.5 | 5 | 0.838 | -5 | 9 |
| EXP2 | 0.906 | -6 | 8 | 0.819 | -14 | 16 |
| EXP3 | 0916 | -7 | 8 | 0.837 | -14 | 16 |
| EXP4 | 0.936 | -11 | 12 | 0.884 | -18 | 20 |
| GLDAS | 0.855 | -6 | 9 | 0.898 | -12 | 14 |
| Mesoscale circulation | | | | | | |
| EXP1 | 0.932 | 6 | 9 | 0.905 | -15 | 15 |
| EXP2 | 0.947 | 0.4 | 5 | 0.944 | -20 | 30 |
| EXP3 | 0.945 | 0.006 | 6 | 0.948 | -20 | 30 |
| EXP4 | 0.941 | -5 | 8 | 0.935 | -30 | 30 |
| GLDAS | 0.940 | -6 | 8 | 0.756 | -30 | 30 |
| Eastern synoptic advection | | | | | | |
| EXP1 | 0.710 | 3 | 13 | 0.963 | -20 | 30 |
| EXP2 | 0.697 | -4 | 11 | 0.962 | -30 | 30 |
| EXP3 | 0.714 | -5 | 12 | 0.968 | -30 | 30 |
| EXP4 | 0.756 | -10 | 13 | 0.977 | -40 | 40 |
| GLDAS | 0.848 | -1.0 | 7 | 0.908 | -19 | 20 |
| Western synoptic advection (Cloudy) | | | | | | |
| EXP1 | 0.587 | 2 | 12 | 0.672 | -0.13 | 14 |
| EXP2 | 0.668 | -6 | 12 | 0.774 | -10 | 15 |
| EXP3 | 0.682 | -7 | 12 | 0.775 | -10 | 15 |
| EXP4 | 0.725 | -12 | 15 | 0.776 | -13 | 18 |
| GLDAS | 0.905 | -0.5 | 5 | 0.965 | -8 | 10 |
| All atmospheric conditions | | | | | | |
| EXP1 | 0.828 | 3 | 9 | 0.829 | -10 | 16 |
| EXP2 | 0.832 | -4 | 9 | 0.847 | -19 | 20 |
| EXP3 | 0.837 | -4 | 9 | 0.858 | -19 | 20 |
| EXP4 | 0.851 | -9 | 12 | 0.866 | -20 | 30 |
| GLDAS | 0.857 | -5 | 8 | 0.782 | -17 | 20 |






**Table 7. Correlation coefficient (R), Mean Bias Error (MBE; W m$^{-2}$) and Root Mean Square Error (RMSE; W m$^{-2}$) for RAMS-EXP3 experiment, SEVIRI and GLDAS shortwave radiation over BON and BRX, considering the distinct dominant atmospheric**
**conditions over the study area.**

| | BON | | | BRX | | |
|---|---|---|---|---|---|---|
| | R | MBE | RMSE | R | MBE | RMSE |
| Western synoptic advection | | | | | | |
| EXP3 | 0.998 | 0.8 | 30 | 0.994 | 20 | 50 |
| SEVIRI | 0.997 | -15 | 30 | 0.987 | 11 | 60 |
| GLDAS | 0.998 | -8 | 30 | 0.991 | 20 | 60 |
| Mesoscale circulation | | | | | | |
| EXP3 | 0.996 | 8 | 40 | 0.992 | 30 | 60 |
| SEVIRI | 0.995 | -6 | 40 | 0.980 | 15 | 80 |
| GLDAS | 0.999 | 2 | 20 | 0.992 | 30 | 60 |
| Eastern synoptic advection | | | | | | |
| EXP3 | 0.984 | 30 | 80 | 0.989 | 50 | 80 |
| SEVIRI | 0.997 | 10 | 30 | 0.987 | 30 | 70 |
| GLDAS | 0.996 | 9 | 40 | 0.990 | 40 | 70 |
| Western synoptic advection (Cloudy) | | | | | | |
| EXP3 | 0.826 | -40 | 200 | 0.624 | -40 | 300 |
| SEVIRI | 0.993 | 4 | 50 | 0.937 | -0.2 | 120 |
| GLDAS | 0.949 | -50 | 140 | 0.773 | 30 | 200 |
| All atmospheric conditions | | | | | | |
| EXP3 | 0.973 | 2 | 90 | 0.956 | 19 | 110 |
| SEVIRI | 0.995 | -6 | 40 | 0.979 | 13 | 80 |
| GLDAS | 0.988 | -9 | 60 | 0.973 | 30 | 90 |







**Table 8. Same as Table 7, but for the longwave radiation (W m$^{-2}$).**

| | BON | | | BRX | | |
|---|---|---|---|---|---|---|
| | R | MBE | RMSE | R | MBE | RMSE |
| Western synoptic advection | | | | | | |
| EXP3 | 0.868 | -30 | 30 | 0.750 | -40 | 40 |
| SEVIRI | 0.916 | 5 | 9 | 0.904 | -6 | 13 |
| GLDAS | 0.713 | -5 | 15 | 0.747 | -17 | 20 |
| Mesoscale circulation | | | | | | |
| EXP3 | 0.837 | -20 | 20 | 0.411 | -40 | 40 |
| SEVIRI | 0.946 | 0.8 | 6 | 0.495 | -20 | 30 |
| GLDAS | 0.969 | -10 | 11 | 0.568 | -30 | 40 |
| Eastern synoptic advection | | | | | | |
| EXP3 | 0.912 | -30 | 30 | 0.604 | -50 | 50 |
| SEVIRI | 0.983 | 4 | 7 | 0.447 | -16 | 30 |
| GLDAS | 0.955 | -9 | 15 | 0.533 | -40 | 40 |
| Western synoptic advection (Cloudy) | | | | | | |
| EXP3 | 0.186 | -1.6 | 30 | 0.326 | -15 | 30 |
| SEVIRI | 0.711 | 14 | 17 | 0.680 | -5 | 18 |
| GLDAS | -0.206 | 1.1 | 20 | 0.682 | -16 | 20 |
| All atmospheric conditions | | | | | | |
| EXP3 | 0.760 | -20 | 30 | 0.743 | -40 | 40 |
| SEVIRI | 0.947 | 5 | 10 | 0.799 | -12 | 20 |
| GLDAS | 0.851 | -6 | 16 | 0.773 | -20 | 30 |





**Table 9. Correlation coefficient (R), Mean Bias Error (MBE; W m$^{-2}$) and Root Mean Square Error (RMSE; W m$^{-2}$) for the RAMS-simulated upward longwave radiation over BRX, considering the distinct dominant atmospheric conditions over the study area.**

| | BRX | | |
|---|---|---|---|
| | R | MBE | RMSE |
| Western synoptic advection | | | |
| EXP1 | 0.949 | 14 | 20 |
| EXP3 | 0.955 | 25 | 30 |
| EXP4 | 0.958 | 30 | 30 |
| Mesoscale circulation | | | |
| EXP1 | 0.968 | 13 | 18 |
| EXP3 | 0.971 | 20 | 20 |
| EXP4 | 0.971 | 30 | 30 |
| Eastern synoptic advection | | | |
| EXP1 | 0.972 | 15 | 17 |
| EXP3 | 0.977 | 20 | 30 |
| EXP4 | 0.982 | 30 | 30 |
| Western synoptic advection (Cloudy) | | | |
| EXP1 | 0.499 | 5 | 20 |
| EXP3 | 0.677 | 19 | 30 |
| EXP4 | 0.694 | 20 | 30 |
| All atmospheric conditions | | | |
| EXP1 | 0.932 | 13 | 20 |
| EXP3 | 0.942 | 20 | 30 |
| EXP4 | 0.944 | 30 | 30 |







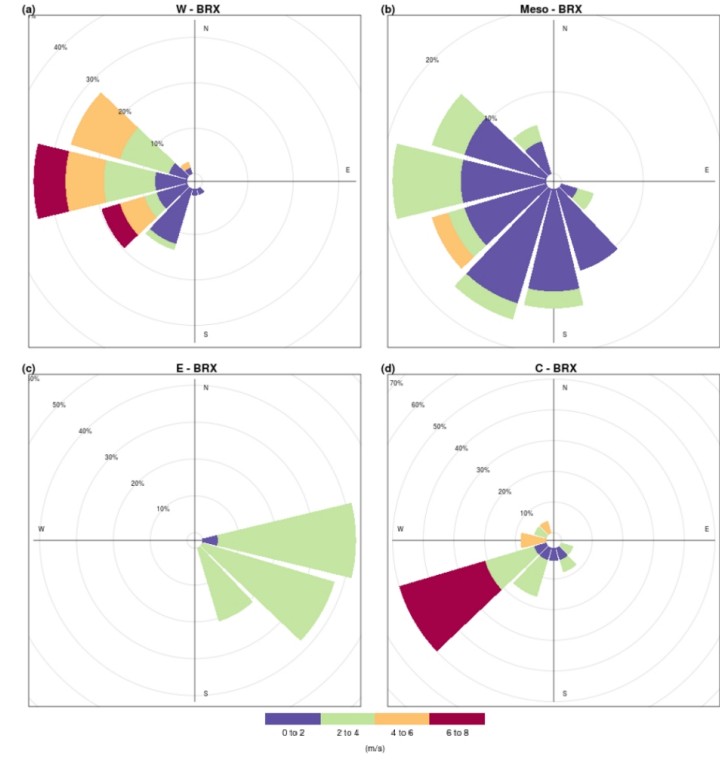

**Figure 1: Observed 10-m wind field (wind direction in º; wind speed in m s$^{-1}$) under the different atmospheric conditions over BRX: Western synoptic advection (W; a), mesoscale circulations (Meso; b), Eastern synoptic advection (E; c) and Western synoptic advection with the presence of cloudiness (C; d).**

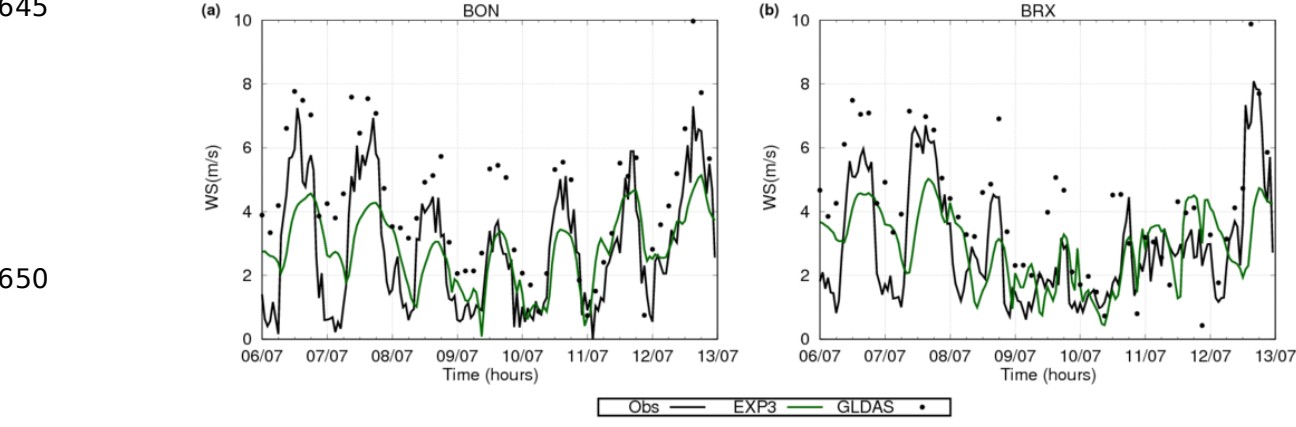

**Figure 2: Observed (black line), EXP3 RAMS simulation (green line) and GLDAS (dot points) 10-m wind speed (m s$^{-1}$) time series over: BON (a) and BRX (b).**





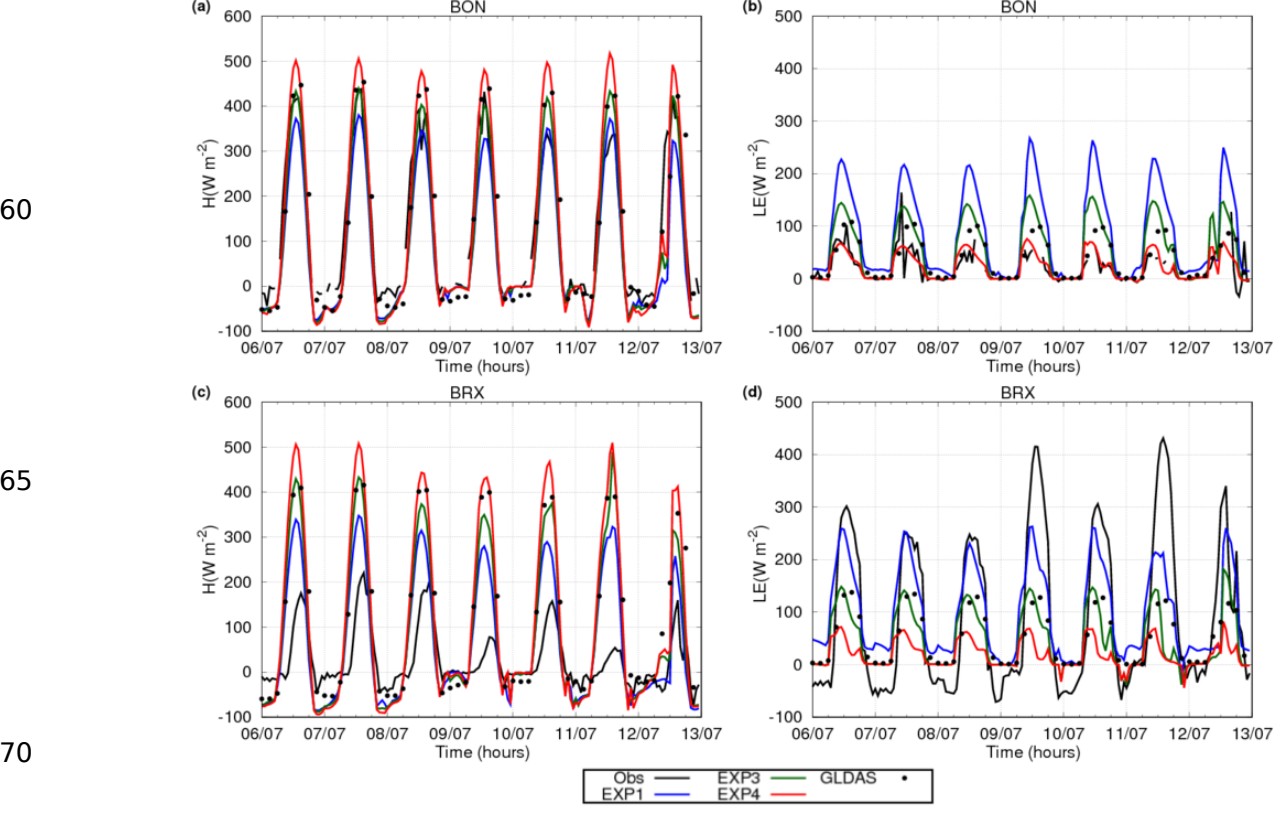

**Figure 3: Observed time series (black line), GLDAS (dot black), and RAMS-simulated surface sensible heat flux (left; W m$^{-2}$) and surface latent heat flux (right; W m$^{-2}$), over BON (a,b) and BRX (c,d).**



**Figure 4: Comparison of RAMS EXP1, EXP3 and EXP4 with EXP2 simulation for the sensible heat flux (a; W m$^{-2}$), latent heat flux (b; W m$^{-2}$), 2-m temperature (c; ºC) and 2-m relative humidity (d; %) over BON weather station.**





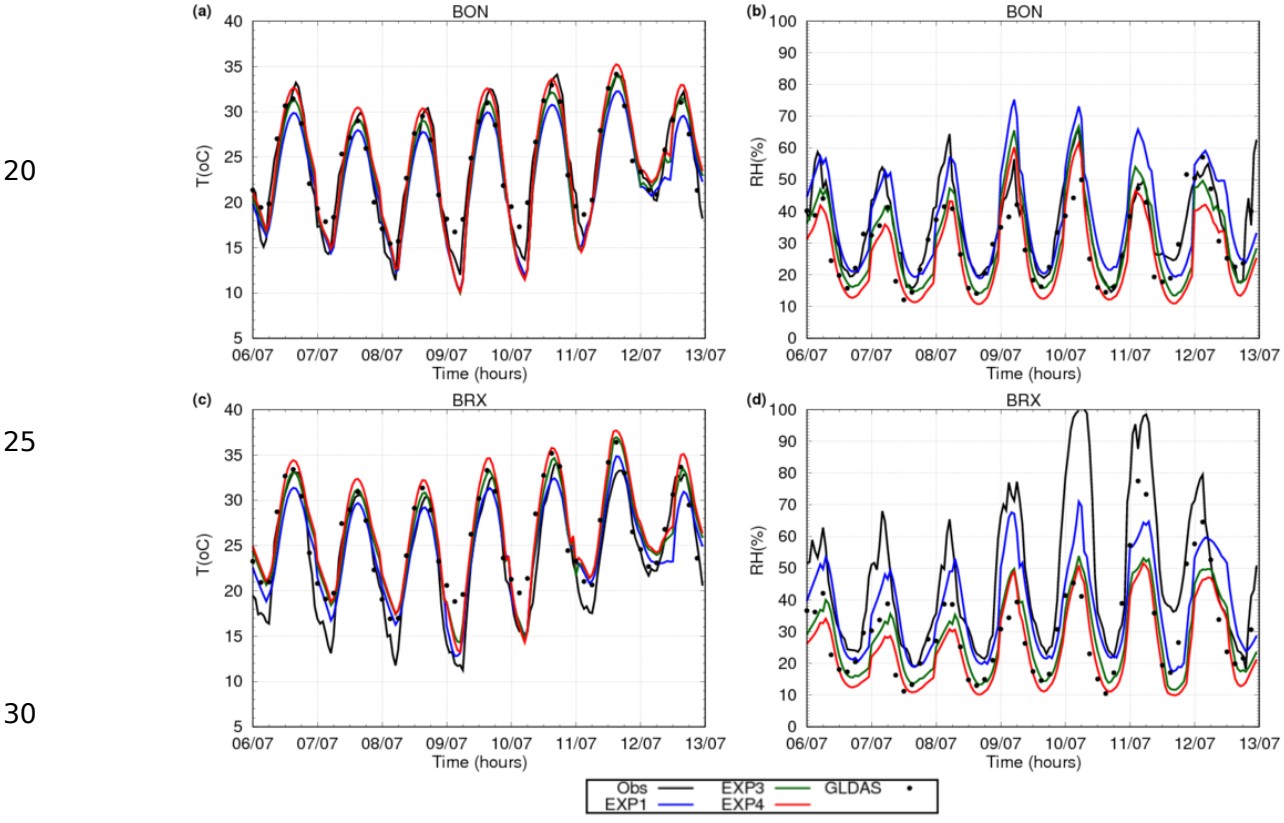

**Figure 5: Same as Fig. 3 but for the 2-m temperature (left; ºC) and the 2-m relative humidity (right; %).**



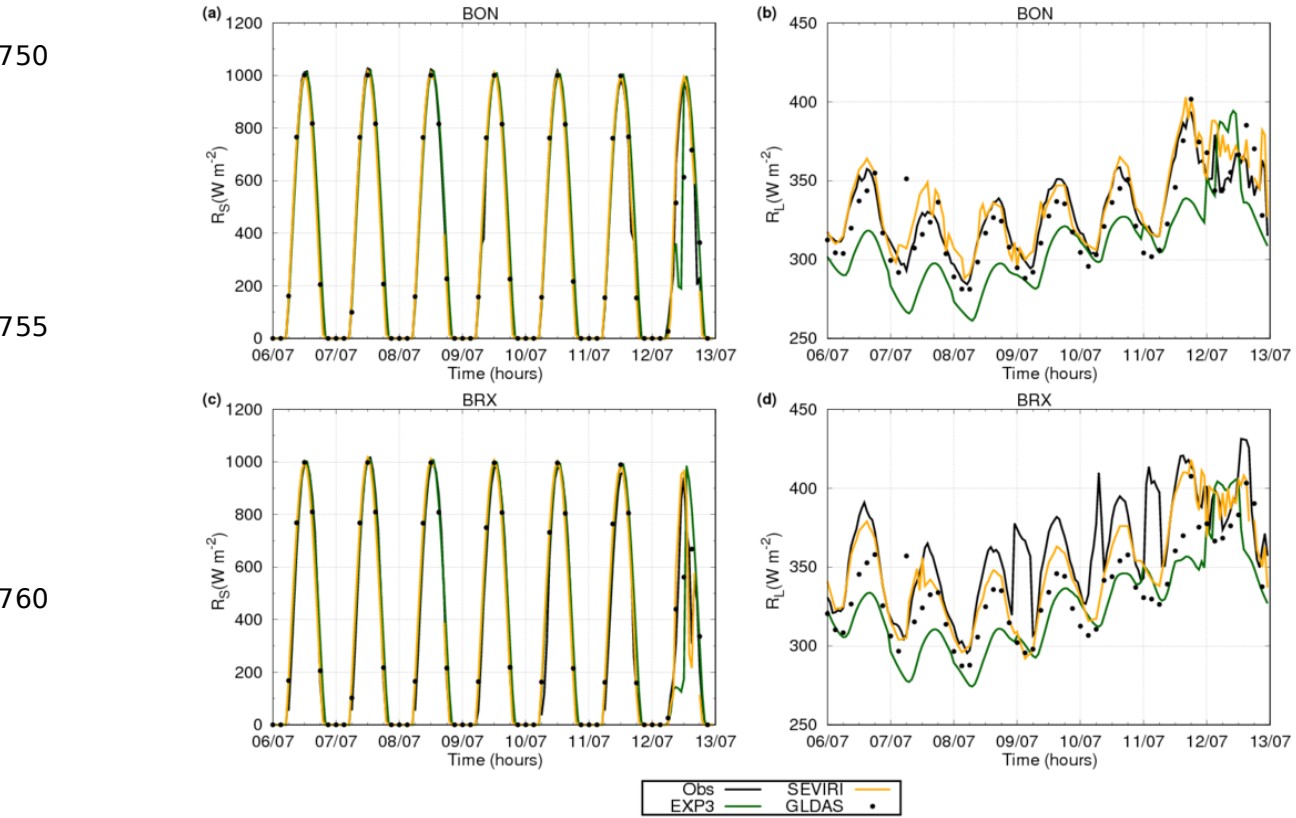

**Figure 6: Observed (black line), EXP3 RAMS simulation (green line), MSG-SEVIRI (orange line) and GLDAS (dot points) incident shortwave radiation (W m$^{-2}$) and incident longwave radiation (W m$^{-2}$) over BON (a,b) and BRX (c,d).**



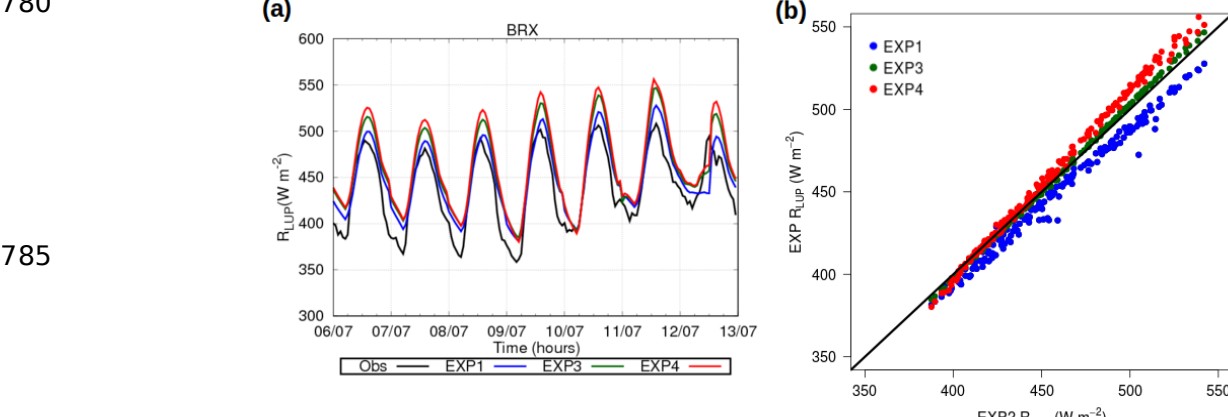

**Figure 7: Comparison of different RAMS experiments with the observed upward longwave radiation (W m$^{-2}$) over BRX (a), and scatterplot of RAMS EXP1, EXP3 and EXP4 *vs.* EXP2 for this magnitude over the same geographical location (b).**





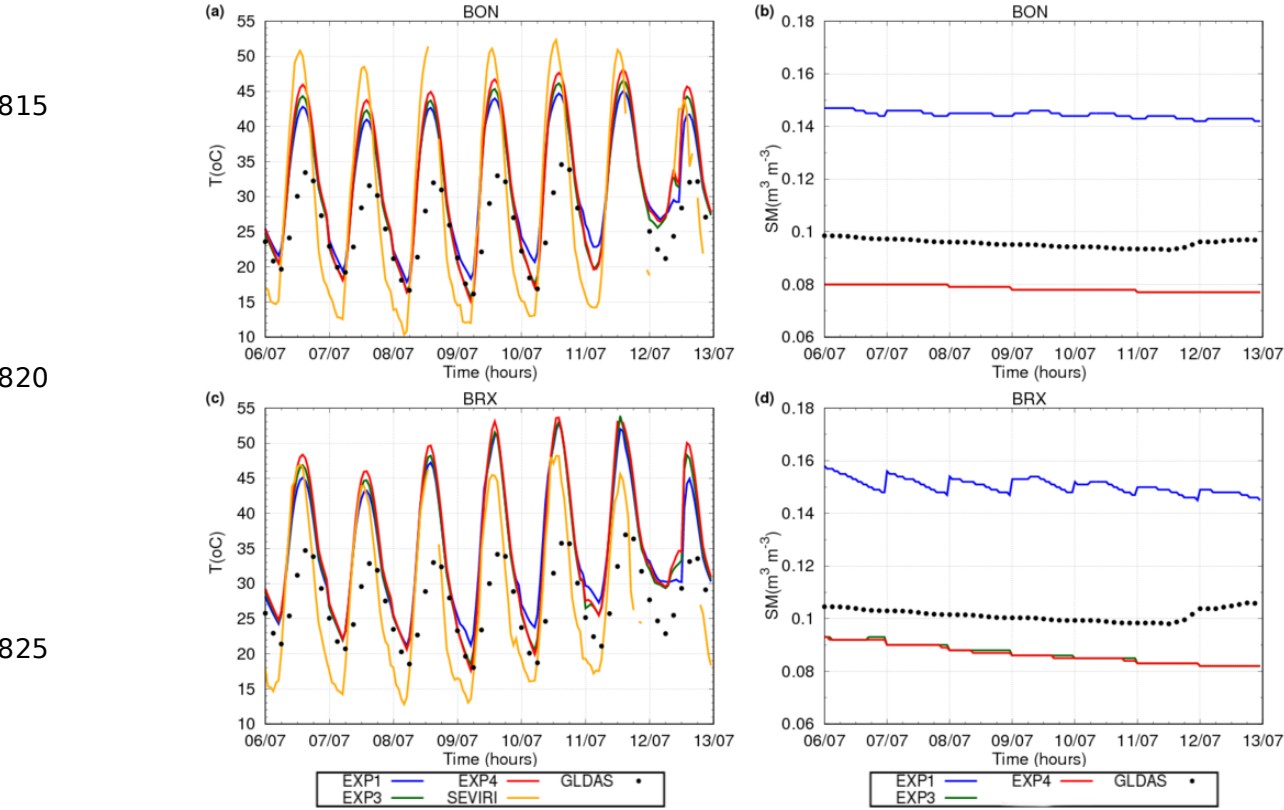

**Figure 8: Comparison of different RAMS experiments with MSG-SEVIRI and GLDAS LST (ºC) over BON (a) and BRX(c), and with GLDAS soil moisture (m³ m⁻³) over these weather stations, (b) and (d), respectively.**





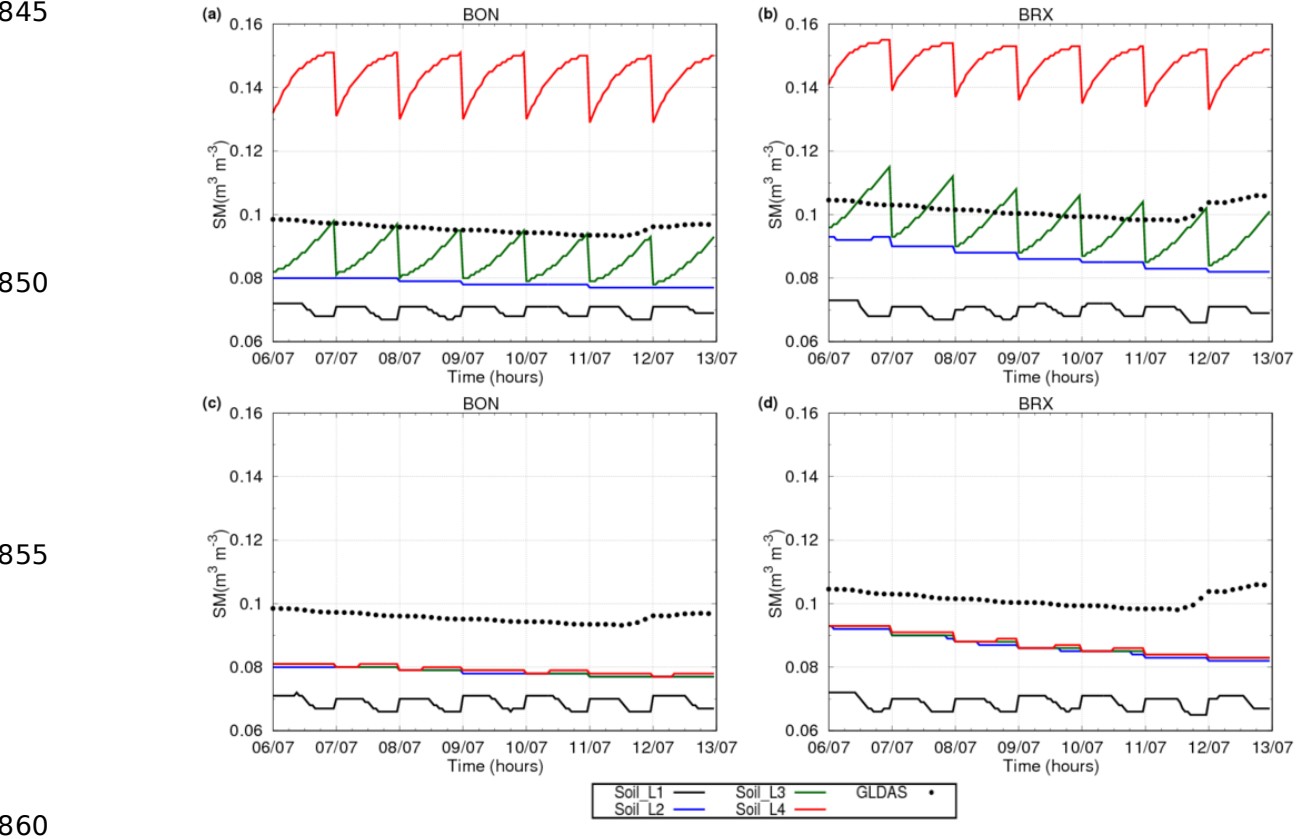

**Figure 9: Comparison of the soil moisture (m³ m⁻³) simulated by RAMS within the upper four soil levels and GLDAS 0-10 cm soil layer. RAMS-EXP3 (upper) over BON (a) and BRX (b). RAMS-EXP4 over BON (c) and BRX (d). RAMS soil levels are represented as: Soil_L1 is located at 2 cm, Soil_L2 at 4.5 cm, Soil_L3 at 7.5 cm and Soil_L4 at 10.5 cm.**