# Peer review of "Improved meteorology and surface energy fluxes in mesoscale modelling using adjusted initial vertical soil moisture profiles"

_Hydrology and Earth System Sciences, 2017_

## Referee Comment (RC1) · Anonymous Referee #1 · 15 Jan 2018

<Overall comments>

This manuscript discusses the sensitivity of soil moisture profile to the estimation of heat fluxes, and their error propagation to latent heat through lower boundary layer. Authors also clearly demonstrated the importance of RAMS atmospheric model initialization.

However, novelty is unclear. Although RAMS model is very interesting to look at, it is already known that soil moisture initialization is important for and and sensitive to atmospheric and land surface model performance. Despite of signifiance, it is our challenge that our models still can't reasonably simulate both surface and subsurface

soil moisture (this paper didn't resolve such an issue or improve a model formulation of soil moisture vertical profile, or provide a better parameterization of soil properties). Some mechanism is obviously missing from current model structures. For this reason, ESA and NASA launched soil moisture satellite to provide soil moisture 'observations'. However, the data are still not perfect so that most of weather prediction centers have satellite data assimilation procedures to improve soil moisture initialization. That's where we are, although there are still several challenges remaning even in their works. Considering our current status, it is unsure how much the message of this paper will help people who are struggling in this field. Please clarify in abstract the novelty of your findings.

Other issues may include:

- Validation methods of RAMS model performance:

I realized that there are no field measurements on soil moisture, and surface fuxes, although these are the key parameters or variables that this manuscript tried to discuss, ultimately. Rather, they borrowed GLDAS data or SEVIRI data as reference. However, there is a huge difference in scale between point-scale flux station, and foot-print scale satellite data at 0.25 degree. 0.25 degree is a quite coarse scale as authors are also aware of at L230. In this case, intercomparisons, consistency or agreements between datasets only provide limited interpretation. Rather than true values or robustness or correctness, it may be argued that it informs of spatial homogeneity. At L230, Authors also admitted that due to a coarse resolution of GLDAS, it is not possible to capture the features at local sites. Please discuss whether flux station data can spatially represent the meso-scale circulations, or synoptic advection in Figure 1, and whether reduced MBE, RMSEs can say a better representation, despite of a large difference in scale or just spatial homogeneity. Agreements with GLDAS data can really prove the correctness or integrity of models. Although GLDAS is one of the best data sets, they are not perfect, as well. You may use:

[Figure]

Chen, B., Black, T.A., Coops, N.C. et al. Boundary-Layer Meteorol (2009) 130: 137. https://doi.org/10.1007/s10546-008-9339-1

Lee, J.H.; Zhao, C.; Kerr, Y. Stochastic Bias Correction and Uncertainty Estimation of Satellite-Retrieved Soil Moisture Products. Remote Sens. 2017, 9, 847

- information on RAMS model structures

Please try to provide the explanation in terms of model structures for each overestimation or underestimation. Although it is sometimes nicely explained about synoptic advection case at L270, it is not fully explained. For example, at L204, differences reached 100 W/m^-2. However, there is no speculation for the reasons in terms of RAMS model structures.

<Detail comments>

L25: As SEVIRI and LSM GLDAS have a coarse resolution, and still contain errors, they are nether "ground" nor 'truth'. It is just reference data.

L32: Please discuss whether a better agreement between model and simulation can guarantee the robustness of models? What if both of data sets do not play a role of 'ground truth'? in fact, both data sets contain unknown errors. You also said that GLDAS makes an overstimation at L189, and L230.

L33: Please highlight in abstract what the novel finding of your works is.

L100-105: Figures only show a week of study period. Does it exclude a warm-up period?

L111: reference run mean control run?

L115: Please inform a spatial resolution of spatial soil moisture to compare RAMS output with GLDAS or station data.

L125: It may be helpful to illustrate RAMS or LEAF model structure for soil moisture

profile to show their current limitations, because your message is that proper representation or simulation of soil moisture profile is important. Equation (1) and (2) do not inform how LE is influenced by soil moisture.

L145: Please provide foot-print scale coverage of your flux stations, as they were compared with GLDAS or satellite or model data. You did that for satellite at L161. Please provide difference in land cover or soil heterogenity between BRX and BON, as you said at L379. Are they upscaled enough so that we can compare it with GLDAS?

Please see Crow, W.T.; Berg, A.A.; Cosh, M.H.; Loew, A.; Mohanty, B.P.; Panciera, R.; de Rosnay, P.; Ryu, D.; Walker, J.P. Upscaling sparse ground-based soil moisture observations for the validation of coarse-resolution satellite soil moisture products. Rev. Geophys. 2012, 50, RG2002

L149: Please provide STSEB model error structure, as you used that as 'ground truth'.

L190: Do you want to discuss possible reasons for such differences?

L228: extend -> extent

L232: Instead of temperature, please specify that like 2m air temperasture. There are so many types of temperatures, e.g. soil temperature, capony temperature, land surface temperature. Instead of 'moisture', please specify that like (subsurface or surface) soil moisture. Atmosphere also has moisture.

L247: differences in maximum temperature.

L258: global mean bias means a spatial and time average?

L271: Please provide references for the speculation of reasons on differences.

L331: same as above at L232.

L336: is that because meso-scale mixing caused spatial homogenity? High RMSE or MBE may be affected by spatial heterogeniety in case of large discrepancy in scales.

L341: Again, please discuss the scale difference between RAMS and GLDAS.

L345: uppermost 4 layers

L349: 'clear' is unclear to readers. Do you mean larger or higher than uppermost? You may provide more references in relation to deeper soil vertical profile. Examples are:

Juglea, S. Juglea, Y. Kerr, A. Mialon, J.-P. Wigneron, E. Lopez-Baeza, A. Cano, A. Albitar, C. Millan-Scheiding, M. Carmen Antolin, S. Delwart Modelling soil moisture at SMOS scale by use of a SVAT model over the Valencia Anchor Station, Hydrology and Earth System Sciences, 14 (2010), pp. 831-846, 10.5194/hess-14-831-2010

Lee, J.H., Pellarin, T., Kerr, Y. (2014). Inversion of soil hydraulic properties from EnKF analysis of SMOS soil moisture over West Africa. Agri. Forestry. Meteo. 05/2014; 188, 76–88, doi: 10.1016/j.agrformet.2013.12.009

Lee et al., 2012, J. Timmermans, Z. Su, M. Mancini, Calibration of aerodynamic roughness over the Tibetan Plateau with Ensemble Kalman Filter analysed heat flux, Hydrology and Earth System Sciences, 16 (2012), pp. 4291-4302, 10.5194/hess-16-4291-2012

Montaldo and Albertson, 2001 N. Montaldo, J.D. Albertson, On the use of the force-restore SVAT model formulation for stratified soils, Journal of Hydrometeorology, 2 (2001), pp. 571-578

Norman et al., 1995. J.M. Norman, W.P. Kustas, K.S. Humes A two-source approach for estimating soil and vegetation energy fluxes in observations of directional radiometric surface temperature, Agricultural and Forest Meteorology, 77 (1995), pp. 263-293

Pollacco and Mohanty, 2012, Uncertainties of water fluxes in soil-vegetation-atmosphere transfer models: Inverting surface soil moisture and evapotranspiration retrieved from remote sensing, Vadose Zone Journal (2012), 10.2136/vzj2011.0167

<Figures>

Figure 1. why there is no BON? Please comment it if there is no advection there?

Figure 6. Did you compare it with EXP 3 simulation, instead of EXP 1 or 2, because GLDAS covers 0-10 cm? Does RAMS read the soil layer specifically at 5 cm or integrate the whole profile ranging from 0 to 10cm?

Figure 8. It is not absolute that LST is inversely proportional with Soil moisture. However, it is commonly found. It's quite interesting that BON site has no relationship between LST and SM. BRX seems to have a positive correlation between LST and SM...? Would you explain that a little bit by using appropriate references?

Figure 9. deeper layer is important, but it is not covered by satellite observation. I wish to pursue authors' suggestion to improve the representatin of deeper layer? That may make a productive discussion. There are uncertainty in subsurface soil properties, as illustrated in the references suggested above.

---

## Referee Comment (RC2) · Anonymous Referee #2 · 5 Apr 2018

In this manuscript, the authors analyse the impact of different soil moisture initialization on meteorological forecasts for two sites in Spain. The forecasting model is the well-established RAMS scheme. The forecasting period is from 6th to 11th of July 2011. The evaluation focuses on turbulent fluxes (i.e., sensible and latent heat), radiation components (i.e., downwards shortwave and upward longwave and shortwave radiation), land surface temperature, and soil moisture. The authors thus provide a comprehensive evaluation of atmosphere-land surface exchange fluxes and land surface states. If local observations of these fluxes are not available, then simulated quantities from GLDAS (based on the Noah LSM) are used. I would like to congratulate the authors to this comprehensive analysis which reveals model deficiencies. Unfor-

tunately, the authors provide little discussion of their results and there are substantial aspects left out of the analysis. My major criticsm is that all of the results are a direct consequence of the thermodynamic physics underlying land surface schemes. If soil moisture is reduced, then sensible heat will increase, latent heat will decrease, land surface temperature and upward longwave radiation will increase. An important aspect here is that biases in turbulent fluxes are reduced, but biases in radiation are increased (i.e. upward longwave radiation). Thus, the authors simply shift biases from one component of the model to another, which is not discussed at all in the manuscript. The model bias could originate from different sources (e.g., model structural errors, errors in model parameters, etc.). The authors only focus on one source, that is the initialization of soil moisture without much motivation or discussion why other sources are left out. It is well known that simulated soil moisture is a quantity that heavily depends on the model structure (Koster et al., 2009). Therefore, I am not surprised that unsatisfying model behaviour is emerging if RAMS is initialized with soil moisture from another modelling system (NCEP FNL). This is especcially true given the large differences in the spatial resolution (3 km vs 1 deg). Moreover, I found seminal work on forecast with RAMS that initialized soil moisture as 50% of field capacity (Castro et al., 2005) that has not been considered in this work. Based on this analysis, I find the findings of the authors rather trivial and a consequence of transferring soil moisture from one model to another without considering differences in model structures. Moreover, the manuscript has frequent references to wrong figures and tables which makes it often difficult to correctly understand what the authors want to express (see further comments below). As a result of my analysis, I recommend to reject this manuscript. The authors need to reassess their strategy for initialization of soil moisture, but this would be a new paper.

Further general comments:

The title is misleading because there is no general improvement but errors are shifted from turbulent fluxes (i.e., sensible and latent heat) to outgoing longwave radiation.

There are frequently general statements that can be made without this study because

these are based on the thermodynamic physics underlying any land-surface scheme. For example, p. 9 l. 275: "Drying the soil,... " and p. 10 l. 287ff: "Considering the meteorological variables..."

I do not think that evaluating relative humidity is a good choice because it depends on both the atmospheric vapour pressure and air temperature. Biases in the later are thus transferred to biases in relative humidity. Using specific humidity would be better because these are more independent of temperature biases.

Throughout the manuscript, the authors are referring to different circulation states (e.g., mesoscale circulation). Results are separated for these in the table. It would have been helpful if the corresponding periods of the different circulations are also highlighted in all figures displaying time series, for example Fig. 8.

Why is Exp. 2 not displayed in Figs. 3 and 5?

Further specific comments:

- p. 7 l. 204: It should read Fig. 3c.

- p. 8 l. 249: It should read "sign"

- p. 9 l. 265: It should read Fig. 5d

- p. 9 l. 276ff.: The differences in thermodynamic variables betweeen Exp. 2 and 3 are not displayed in table 7. In table 4, the differences between Exp. 2 and 3 are as large as those between the other experiments. This statement is thus misleading. I also would avoid the use of the word "really" in general.

- p. 10 l. 288: Fig. 6 has not been discussed yet in the text. Why is it referenced here?

- p. 10 l. 294f: Soil moisture is controlling both the partitioning of energy into sensible heat and latent heat and is a model dependent quantity. However, changing soil moisture to reduce a cold bias in temperature is just one option. The cold bias could also originate from deficiencies in process parametrization including model parameters. These are often conceptual and associated with substantial uncertainties (for example the exponents in equation 2). The authors should answer why the initial soil moisture field they use is too wet and justify the use of a drier one.

- p. 10 l. 304f: I do not understand the use of the word "adjust" here.

- p. 10 l. 306f: According to table 7, I would argue that all of these datasets provide a comparable performance.

- p. 10 l. 323f: The large bias by RAMS in downward longwave radiation could also lead to the cold bias the authors try to remove by adjusting soil moisture. This gives the impression that the authors want to get the right result for the wrong reason. The authors should perform further evaluation of cloudiness and the atmospheric radiation scheme to identify the origin of this bias.

- p. 11 l. 326f: It is surprising to see that upward longwave radiation is overestimated. This clearly indicates that there is a problem in the model structure because removing biases in the turbulent fluxes (i.e., sensible and latent heat) introduce biases in this radiation component. This indicates that surface and skin temperature cannot be calculated in a way to satisfy both turbulent fluxes and radiation.

- p. 11 l. 336ff: I cannot see a reversed trend between the derived LST from RAMS and SEVIRI at the BON station during daytime. The authors need to clarify what they want to express here.

- p. 11 l. 338ff: I do not see an added value of the evaluation of soil moisture because of two reasons. 1) It is not a result that EXP3 and EXP4 have drier soils than EXP1 because this is how the experiments have been designed! 2) The comparison against GLDAS is troublesome because GLDAS is run at a much coarser resolution and also has a different soil depth in the first layer compared to RAMS (10cm vs. 2cm).

- p. 11 l. 345ff: I am confused about the presented results because I would have expected that the black solid lines in Figure 9 should be equal to the corresponding red

and green lines in Figure 8b and 8d. Both of these are claimed to show soil moisture in the top layer, but they are different. The striking feature of Figure 9a and Figure 9b is that the soil in Layer 3 and 4 becomes wetter with lead time. I assume that this water is transferred from lower soil layers by capillary rise but the rate seems to be very high.

- p. 12 l. 362: The abbreviation BRX should be reintroduced here.

- p. 12 l. 365: The authors did not show that any observations of soil moisture. This conclusion is not supported by the manuscript. GLDAS is used as a reference, but values between GLDAS and RAMS cannot be compared because of substantial differences in spatial and vertical resolution.

- Conclusions: The authors do not mention at all that the bias in upward longwave radiation is increasing with drier soils and thus, biases are transferred from the turbulent fluxes to the radiation components.

- Figure 5: The x-axis label with Time(hours) is confusing. It should be days. Do not use red and green lines in the same plot because this is not color-blind friendly.

References:

Christopher L Castro et al. "Dynamical downscaling: Assessment of value retained and added using the Regional Atmospheric Modeling System (RAMS)." J Geophys Res, 2005 vol. 110 (D5) pp. 681-21. http://doi.wiley.com/10.1029/2004JD004721

Randal D Koster, Zhichang Guo, Rongqian Yang, Paul A Dirmeyer, Kenneth Mitchell, and Michael J Puma. "On the Nature of Soil Moisture in Land Surface Models." J Climate, 2009 vol. 22 (16) pp. 4322-4335. http://journals.ametsoc.org/doi/abs/10.1175/2009JCLI2832.1